# CapProNet: Deep Feature Learning via Orthogonal Projections onto Capsule Subspaces

**Liheng Zhang**[†]**, Marzieh Edraki**[†]**, and Guo-Jun Qi**[†‡*]

[†]Laboratory for **MA**chine **P**erception and **LE**arning,
University of Central Florida
`http://maple.cs.ucf.edu`

[‡]Huawei Cloud, Seattle, USA

## Abstract

In this paper, we formalize the idea behind capsule nets of using a capsule vector rather than a neuron activation to predict the label of samples. To this end, we propose to learn a group of capsule subspaces onto which an input feature vector is projected. Then the lengths of resultant capsules are used to score the probability of belonging to different classes. We train such a Capsule Projection Network (CapProNet) by learning an orthogonal projection matrix for each capsule subspace, and show that each capsule subspace is updated until it contains input feature vectors corresponding to the associated class. We will also show that the capsule projection can be viewed as normalizing the multiple columns of the weight matrix simultaneously to form an orthogonal basis, which makes it more effective in incorporating novel components of input features to update capsule representations. In other words, the capsule projection can be viewed as a multi-dimensional weight normalization in capsule subspaces, where the conventional weight normalization is simply a special case of the capsule projection onto 1D lines. Only a small negligible computing overhead is incurred to train the network in low-dimensional capsule subspaces or through an alternative hyper-power iteration to estimate the normalization matrix. Experiment results on image datasets show the presented model can greatly improve the performance of the state-of-the-art ResNet backbones by $10 - 20\%$ and that of the Densenet by $5 - 7\%$ respectively at the same level of computing and memory expenses. The CapProNet establishes the competitive state-of-the-art performance for the family of capsule nets by significantly reducing test errors on the benchmark datasets.

## 1 Introduction

Since the idea of capsule net [15, 9] was proposed, many efforts [8, 17, 14, 1] have been made to seek better capsule architectures as the next generation of deep network structures. Among them are the dynamic routing [15] that can dynamically connect the neurons between two consecutive layers based on their output capsule vectors. While these efforts have greatly revolutionized the idea of building a new generation of deep networks, there are still a large room to improve the state of the art for capsule nets.

In this paper, we do not intend to introduce some brand new architectures for capsule nets. Instead, we focus on formalizing the principled idea of using the overall length of a capsule rather than

---

[*]Corresponding author: G.-J. Qi, email: guojunq@gmail.com and guojun.qi@huawei.com.

a single neuron activation to model the presence of an entity [15, 9]. Unlike the existing idea in literature [15, 9], we formulate this idea by learning a group of *capsule subspaces* to represent a set of entity classes. Once capsule subspaces are learned, we can obtain set of capsules by performing an orthogonal projection of feature vectors onto these capsule subspaces.

Then, one can adopt the *principle of separating* the presence of an entity and its instantiation parameters into capsule length and orientation, respectively. In particular, we use the lengths of capsules to score the presence of entity classes corresponding to different subspaces, while their orientations are used to instantiate the parameters of entity properties such as poses, scales, deformations and textures. In this way, one can use the capsule length to achieve the intra-class *invariance* in detecting the presence of an entity against appearance variations, as well as model the *equivalence* of the instantiation parameters of entities by encoding them into capsule orientations [15].

Formally, each capsule subspace is spanned by a basis from the columns of a weight matrix in the neural network. A capsule projection is performed by projecting input feature vectors fed from a backbone network onto the capsule subspace. Specifically, an input feature vector is orthogonally decomposed into the capsule component as the projection onto a capsule subspace and the complement component perpendicular to the subspace. By analyzing the gradient through the capsule projection, one can show that a capsule subspace is iteratively updated along the complement component that contains the *novel* characteristics of the input feature vector. The training process will continue until all presented feature vectors of an associated class are well contained by the corresponding capsule subspace, or simply the back-propagated error accounting for misclassification caused by capsule lengths vanishes.

We call the proposed deep network with the capsule projections the *CapProNet* for brevity. The CapProNet is friendly to any existing network architecture – it is built upon the embedded features generated by some neural networks and outputs the projected capsule vectors in the subspaces according to different classes. This makes it amenable to be used together with existing network architectures. We will conduct experiments on image datasets to demonstrate the CapProNet can greatly improve the state-of-the-art results by sophisticated networks with only small negligible computing overhead.

## 1.1 Our Findings

Briefly, we summarize our main findings from experiments upfront about the proposed CapProNet.

- The proposed CapProNet significantly advances the capsule net performance [15] by reducing its test error from $10.3\%$ and $4.3\%$ on CIFAR10 and SVHN to $3.64\%$ and $1.54\%$ respectively upon the chosen backbones.

- The proposed CapProNet can also greatly reduce the error rate of various backbone networks by adding capsule projection layers into these networks. For example, The error rate can be reduced by more than $10 - 20\%$ based on Resnet backbone, and by more than $5 - 6\%$ based on densenet backbone, with only $< 1\%$ and $0.04\%$ computing and memory overhead in training the model compared with the backbones.

- The orthogonal projection onto capsule subspaces plays a critical role in delivering competitive performance. On the contrary, simply grouping neurons into capsules could not obviously improve the performance. This shows the capsule projection plays an indispensable role in the CapProNet delivering competitive results.

- Our insight into the gradient of capsule projection in Section 2.3 explains the advantage of updating capsule subspaces to continuously contain novel components of training examples until they are correctly classified. We also find that the capsule projection can be viewed as a high-dimensional extension of weight normalization in Section 2.4, where the conventional weight normalization is merely a simple case of the capsule projection onto 1D lines.

The source code is available at `https://github.com/maple-research-lab`.

The remainder of this paper is organized as follows. We present the idea of the Capsule Projection Net (CapProNet) in Section 2, and discuss the implementation details in Section 3. The review of related work follows in Section 4, and the experiment results are demonstrated in Section 5. Finally, we conclude the paper and discuss the future work in Section 6.

## 2 The Capsule Projection Nets

In this section, we begin by shortly revisiting the idea of conventional neural networks in classification tasks. Then we formally present the orthogonal projection of input feature vectors onto multiple capsule subspaces where capsule lengths are separated from their orientations to score the presence of entities belonging to different classes. Finally, we analyze the gradient of the resultant capsule projection by showing how capsule subspaces are updated iteratively to adopt novel characteristics of input feature vectors through back-propagation.

### 2.1 Revisit: Conventional Neural Networks

Consider a feature vector $\mathbf{x} \in \mathbb{R}^d$ generated by a deep network to represent an input entity. Given its ground truth label $y \in \{1, 2, \cdots, L\}$, the output layer of the deep network aims to learn a group of weight vectors $\{\mathbf{w}_1, \mathbf{w}_2, \cdots, \mathbf{w}_L\}$ such that

$$\mathbf{w}_y^T \mathbf{x} > \mathbf{w}_l^T \mathbf{x}, \text{ for all}, l \neq y. \tag{1}$$

This hard constraint is usually relaxed to a differentiable softmax objective, and the backpropagation algorithm is performed to train $\{\mathbf{w}_1, \mathbf{w}_2, \cdots, \mathbf{w}_L\}$ and the backbone network generating the input feature vector $\mathbf{x}$.

### 2.2 Capsule Projection onto Subspaces

Unlike simply grouping neurons to form capsules for classification, we propose to learn a group of capsule subspaces $\{\mathcal{S}_1, \mathcal{S}_2, \cdots, \mathcal{S}_L\}$, each associated with one of $L$ classes. Suppose we have a feature vector $\mathbf{x} \in \mathbb{R}^d$ generated by a backbone network from an input sample. Then, to learn a proper feature representation, we project $\mathbf{x}$ onto these capsule subspaces, yielding $L$ capsules $\{\mathbf{v}_1, \mathbf{v}_2, \cdots, \mathbf{v}_L\}$ as projections. Then, we will use the lengths of these capsules to score the probability of the input sample belonging to different classes by assigning it to the one according to the longest capsule.

Formally, for each capsule subspace $\mathcal{S}_l$ of dimension $c$, we learn a weight matrix $\mathbf{W}_l \in \mathbb{R}^{d \times c}$ the columns of which form the basis of the subspace, i.e., $\mathcal{S}_l = \text{span}(\mathbf{W}_l)$ is spanned by the column vectors. Then the orthogonal projection $\mathbf{v}_l$ of a vector $\mathbf{x}$ onto $\mathcal{S}_l$ is found by solving $\mathbf{v}_l = \arg\min_{\mathbf{v} \in \text{span}(\mathbf{W}_l)} \|\mathbf{x} - \mathbf{v}\|_2$. This orthogonal projection problem has the following closed-form solution

$$\mathbf{v}_l = \mathbf{P}_l \mathbf{x}, \text{ and } \mathbf{P}_l = \mathbf{W}_l \mathbf{W}_l^+$$

where $\mathbf{P}_l$ is called projection matrix [2] for capsule subspace $\mathcal{S}_l$, and $\mathbf{W}_l^+$ is the Moore-Penrose pseudoinverse [4].

When the columns of $\mathbf{W}_l$ are independent, $\mathbf{W}_l^+$ becomes $(\mathbf{W}_l^T \mathbf{W}_l)^{-1} \mathbf{W}_l^T$. In this case, since we only need the capsule length $\|\mathbf{v}_l\|_2$ to predict the class of an entity, we have

$$\|\mathbf{v}_l\|_2 = \sqrt{\mathbf{v}_l^T \mathbf{v}_l} = \sqrt{\mathbf{x}^T \mathbf{P}_l^T \mathbf{P}_l \mathbf{x}} = \sqrt{\mathbf{x}^T \mathbf{W}_l \mathbf{\Sigma}_l \mathbf{W}_l^T \mathbf{x}} \tag{2}$$

where $\mathbf{\Sigma}_l = (\mathbf{W}_l^T \mathbf{W}_l)^{-1}$ can be seen as a normalization matrix applied to the transformed feature vector $\mathbf{W}_l^T \mathbf{x}$ as a way to normalize the $\mathbf{W}_l$-transformation based on the capsule projection. As we will discuss in the next subsection, this normalization plays a critical role in updating $\mathbf{W}_l$ along the orthogonal direction of the subspace so that novel components pertaining to the properties of input entities can be gradually updated to the subspace.

In practice, since $c \ll d$, the $c$ columns of $\mathbf{W}_l$ are usually independent in a high-dimensional $d$-D space. Otherwise, one can always add a small $\epsilon \mathbf{I}$ to $\mathbf{W}_l^T \mathbf{W}_l$ to avoid the numeric singularity when taking the matrix inverse. Later on, we will discuss a fast iterative algorithm to compute the matrix inverse with a hyper-power sequence that can be seamlessly integrated with the back-propagation iterations.

## 2.3 Insight into Gradients

In this section, we take a look at the gradient used to update $\mathbf{W}_l$ in each iteration, which can give us some insight into how the CapProNet works in learning the capsule subspaces.

Suppose we minimize a loss function $\ell$ to train the capsule projection and the network. For simplicity, we only consider a single sample $\mathbf{x}$ and its capsule $\mathbf{v}_l$. Then by the chain rule and the differential of inverse matrix [13], we have the following gradient of $\ell$ wrt $\mathbf{W}_l$

$$\frac{\partial \ell}{\partial \mathbf{W}_l} = \frac{\partial \ell}{\partial \|\mathbf{v}_l\|_2} \frac{\partial \|\mathbf{v}_l\|}{\partial \mathbf{W}_l} = \frac{\partial \ell}{\partial \|\mathbf{v}_l\|_2} \frac{(\mathbf{I} - \mathbf{P}_l)\mathbf{x}\mathbf{x}^T \mathbf{W}_l^{+T}}{\|\mathbf{v}_l\|_2} \tag{3}$$

where the operator $(\mathbf{I} - \mathbf{P}_l)$ can be viewed as the projection onto the orthogonal complement of the capsule subspace spanned by the columns of $\mathbf{W}_l$, $\mathbf{W}_l^{+T}$ denotes the transpose of $\mathbf{W}_l^+$, and the factor $\frac{\partial \ell}{\partial \|\mathbf{v}_l\|_2}$ is the back-propagated error accounting for misclassification caused by $\|\mathbf{v}_l\|_2$.

Denote by $\mathbf{x}^\perp \triangleq (\mathbf{I} - \mathbf{P}_l)\mathbf{x}$ the projection of $\mathbf{x}$ onto the orthogonal component perpendicular to the current capsule subspace $\mathcal{S}_l$. Then, the above gradient $\frac{\partial \ell}{\partial \mathbf{W}_l}$ only contains the columns parallel to $\mathbf{x}^\perp$ (up to coefficients in the vector $\mathbf{x}^T \mathbf{W}_l^{+T}$). This shows that the basis of the current capsule subspace $\mathcal{S}_l$ in the columns of $\mathbf{W}_l$ is updated along this orthogonal component of the input $\mathbf{x}$ to the subspace.

One can regard $\mathbf{x}^\perp$ as representing the novel component of $\mathbf{x}$ not yet contained in the current $\mathcal{S}_l$, it shows capsule subspaces are updated to contain the novel component of each input feature vector until all training feature vectors are well contained in these subspaces, or the back-propagated errors vanish that account for misclassification caused by $\|\mathbf{v}_l\|_2$.

Figure 1 illustrates an example of 2-D capsule subspace $\mathcal{S}$ spanned by two basis vectors $\mathbf{w}_1$ and $\mathbf{w}_2$. An input feature vector $\mathbf{x}$ is decomposed into the capsule projection $\mathbf{v}$ onto $\mathcal{S}$ and an orthogonal complement $\mathbf{x}^\perp$ perpendicular to the subspace. In one training iteration, two basis vectors $\mathbf{w}_1$ and $\mathbf{w}_2$ are updated to $\mathbf{w}'_1$ and $\mathbf{w}'_2$ along the orthogonal direction $\mathbf{x}^\perp$, where $\mathbf{x}^\perp$ is viewed as containing novel characteristics of an entity not yet contained by $\mathcal{S}$.

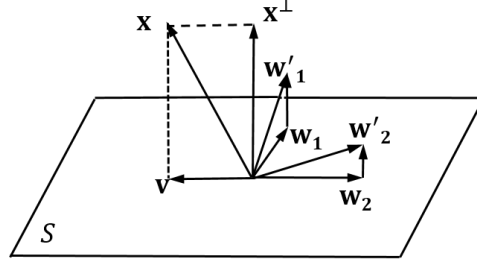

Figure 1: This figure illustrates a 2-D capsule subspace $\mathcal{S}$ spanned by two basis vectors $\mathbf{w}_1$ and $\mathbf{w}_2$. An input feature vector $\mathbf{x}$ is decomposed into the capsule projection $\mathbf{v}$ onto $\mathcal{S}$ and an orthogonal complement $\mathbf{x}^\perp$ perpendicular to the subspace. In one training iteration, two basis vectors $\mathbf{w}_1$ and $\mathbf{w}_2$ are updated to $\mathbf{w}'_1$ and $\mathbf{w}'_2$ along the orthogonal direction $\mathbf{x}^\perp$, where $\mathbf{x}^\perp$ is viewed as containing novel characteristics of an entity not yet contained by $\mathcal{S}$.

## 2.4 A Perspective of Multiple-Dimensional Weight Normalization

As discussed in the last subsection and Figure 2, we can explain the orthogonal components represent the novel information in input data, and the orthogonal decomposition thus enables us to update capsule subspaces by more effectively incorporating novel characteristics/components than the classic capsule nets.

One can also view the capsule projection as normalizing the column basis of weight matrix $\mathbf{W}_l$ simultaneously in a high-dimensional capsule space. If the capsule dimension $c$ is set to 1, it is not hard to see that Eq. (2) can be rewritten by setting $\mathbf{v}_l$ to $\frac{|\mathbf{W}_l^T \mathbf{x}|}{\|\mathbf{W}_l\|}$. It produces the conventional weight normalization of the vector $\mathbf{W}_l \in \mathbb{R}^d$, as a special $1D$ case of the capsule projection. As the capsule dimension $c$ grows, $\mathbf{W}_l$ can be normalized by replacing $\mathbf{v}_l$ with $\mathbf{\Sigma}_l^{1/2} \mathbf{W}_l^T \mathbf{x}$, which keeps $\|\mathbf{v}_l\|$ unchanged in Eq. (2). This enables us to extend the conventional weight normalization to high dimensional capsule subspaces.

## 3 Implementation Details

We will discuss some implementation details in this section, including 1) the computing cost to perform capsule projection and a fast iterative method by using hyper-power sequences without restart; 2) the objective to train the capsule projection.

### 3.1 Computing Normalization Matrix

Taking a matrix inverse to get the normalization matrix $\boldsymbol{\Sigma}_l$ would be expensive with an increasing dimension $c$. But after the model is trained, it is fixed in the inference with only one-time computing. Fortunately, the dimension $c$ of a capsule subspace is usually much smaller than the feature dimension $d$ that is usually hundreds and even thousands. For example, $c$ is typically no larger than $8$ in experiments. Thus, taking a matrix inverse to compute these normalization matrices only incurs a small negligible computing overhead compared with the training of many other layers in a deep network.

Alternatively, one can take advantage of an iterative algorithm to compute the normalization matrix. We consider the following hyper-power sequence

$$\boldsymbol{\Sigma}_l \leftarrow 2\boldsymbol{\Sigma}_l - \boldsymbol{\Sigma}_l \mathbf{W}_l^T \mathbf{W}_l \boldsymbol{\Sigma}_l$$

which has proven to converge to $(\mathbf{W}^T\mathbf{W})^{-1}$ with a proper initial point [2, 3]. In stochastic gradient method, since only a small change is made to update $\mathbf{W}_l$ in each training iteration, thus it is often sufficient to use this recursion to make an one-step update on the normalization matrix from the last iteration. The normalization matrix $\boldsymbol{\Sigma}_l$ can be initialized to $(\mathbf{W}_l^T\mathbf{W}_l)^{-1}$ at the very first iteration to give an ideal start. This could further save computing cost in training the network.

In experiments, a very small computing overhead was incurred in the capsule projection. For example, training the ResNet110 on CIFAR10/100 costed about $0.16$ seconds per iteration on a batch of $128$ images. In comparison, training the CapProNet with a ResNet110 backbone in an end-to-end fashion only costed an additional $< 0.001$ seconds per iteration, that is less than $1\%$ computing overhead for the CapProNet compared with its backbone. For the inference, we did not find any noticeable computing overhead for the CapProNet compared with its backbone network.

### 3.2 Training Capsule Projections

Given a group of capsule vectors $\{\mathbf{v}_1, \mathbf{v}_2, \cdots, \mathbf{v}_L\}$ corresponding to a feature vector $\mathbf{x}$ and its ground truth label $y$, we train the model by requiring

$$\|\mathbf{v}_y\|_2 > \|\mathbf{v}_l\|_2, \text{ for all}, l \neq y.$$

In other words, we require $\|\mathbf{v}_y\|_2$ should be larger than all the length of the other capsules. As a consequence, we can minimize the following negative logarithmic softmax function $\ell(\mathbf{x}, y) = -\log \frac{\exp(\|\mathbf{v}_y\|_2)}{\sum_{l=1}^{L} \exp(\|\mathbf{v}_l\|_2)}$ to train the capsule subspaces and the network generating $\mathbf{x}$ through back-propagation in an end-to-end fashion. Once the model is trained, we will classify a test sample into the class with the longest capsule.

## 4 Related Work

The presented CapProNets are inspired by the CapsuleNets by adopting the idea of using a capsule vector rather than a neural activation output to predict the presence of an entity and its properties [15, 9]. In particular, the overall length of a capsule vector is used to represent the existence of the entity and its direction instantiates the properties of the entity. We formalize this idea in this paper by explicitly learning a group of capsule subspaces and project embedded features onto these subspaces.

The advantage of these capsule subspaces is their directions can represent characteristics of an entity, which contains much richer information, such as its positions, orientations, scales and textures, than a single activation output. By performing an orthogonal projection of an input feature vector onto a capsule subspace, one can find the best direction revealing these properties. Otherwise, the entity is thought of being absent as the projection vanishes when the input feature vector is nearly perpendicular to the capsule subspace.

## 5 Experiments

We conduct experiments on benchmark datasets to evaluate the proposed CapProNet compared with the other deep network models.

### 5.1 Datasets

We use both CIFAR and SVHN datasets in experiments to evaluate the performance.

**CIFAR** The CIFAR dataset contains $50,000$ and $10,000$ images of $32 \times 32$ pixels for the training and test sets respectively. A standard data augmentation is adopted with horizonal flipping and shifting. The images are labeled with 10 and 100 categories, namely CIFAR10 and CIFAR100 datasets. A separate validation set of $5,000$ images are split from the training set to choose the model hyperparameters, and the final test errors are reported with the chosen hyperparameters by training the model on all $50,000$ training images.

**SVHN** The Street View House Number (SVHN) dataset has $73,257$ and $26,032$ images of colored digits in the training and test sets, with an additional $531,131$ training images available. Following the widely used evaluation protocol in literature [5, 11, 12, 16], all the training examples are used without data augmentation, while a separate validation set of $6,000$ images is split from the training set. The model with the smallest validation error is selected and the error rate is reported.

**ImageNet** The ImageNet data-set consists of 1.2 million training and 50k validation images. We apply mean image subtraction as the only pre-processing step on images and use random cropping, scaling and horizontal flipping for data augmentation [6]. The final resolution of both train and validation sets is $224 \times 224$, and 20k images are chosen randomly from training set for tuning hyper parameters.

### 5.2 Backbone Networks

We test various networks such as ResNet [6], ResNet (pre-activation) [7], WideResNet [18] and Densenet [10] as the backbones in experiments. The last output layer of a backbone network is replaced by the capsule projection, where the feature vector from the second last layer of the backbone is projected onto multiple capsule subspaces.

The CapProNet is trained from the scratch in an end-to-end fashion on the given training set. For the sake of fair comparison, the strategies used to train the respective backbones [6, 7, 18], such as the learning rate schedule, parameter initialization, and the stochastic optimization solver, are adopted to train the CapProNet. We will denote the CapProNet with a backbone X by CapProNet+X below.

### 5.3 Results

We perform experiments with various networks as backbones for comparison with the proposed CapProNet. In particular, we consider three variants of ResNets – the classic one reported in [11] with 110 layers, the ResNet with pre-activation [7] with 164 layers, and two paradigms of WideResNets [18] with 16 and 28 layers, as well as densenet-BC [10] with 100 layers. Compared with ResNet and ResNet with pre-activation, WideResNet has fewer but wider layers that reaches smaller error rates as shown in Table 1. We test the CapProNet+X with these different backbone networks to evaluate if it can consistently improve these state-of-the-art backbones. It is clear from Table 1 that the CapProNet+X outperforms the corresponding backbone networks by a remarkable margin. For example, the CapProNet+ResNet reduces the error rate by $19\%$, $17.5\%$ and $10\%$ on CIFAR10, CIFAR100 and SVHN, while CapProNet+Densenet reduces the error rate by $5.8\%$, $4.8\%$ and $6.8\%$ respectively. Finally, we note that the CapProNet significantly advances the capsule net performance [15] by reducing its test error from $10.3\%$ and $4.3\%$ on CIFAR10 and SVHN to $3.64\%$ and $1.54\%$ respectively based on the chosen backbones.

We also evaluate the CapProNet with Resnet50 and Resnet101 backbones for single crop Top-1/Top-5 results on ImageNet validation set. To ensure fair comparison, we retrain the backbone networks based on the offical Resnet model[3], where both original Resnet[6] and CapProNet are trained with the same training strategies on four GPUs. The results are reported in Table 2, where CapProNet+X

Table 1: Error rates on CIFAR10, CIFAR100, and SVHN. The best results are highlighted in bold for the methods with the same network architectures. Not all results on different combinations of network backbones or datasets have been reported in literature, and missing results are remarked "-" in the table.

| Method | Depth | Params | CIFAR10 | CIFAR100 | SVHN |
|---|---|---|---|---|---|
| ResNet [6] | 110 | 1.7M | 6.61 | - | - |
| ResNet (reported by [11]) | 110 | 1.7M | 6.41 | 27.22 | 2.01 |
| CapProNet+ResNet (c=2) | 110 | 1.7M | 5.24 | 22.65 | **1.79** |
| CapProNet+ResNet (c=4) | 110 | 1.7M | 5.27 | 22.45 | 1.82 |
| CapProNet+ResNet (c=8) | 110 | 1.7M | **5.19** | **21.93** | **1.79** |
| ResNet (pre-activation) [7] | 164 | 1.7M | 5.46 | 24.33 | - |
| CapProNet+ResNet (pre-activation c=4) | 164 | 1.7M | 4.88 | 21.37 | - |
| CapProNet+ResNet (pre-activation c=8) | 164 | 1.7M | 4.89 | **20.91** | - |
| WideResNet [18] | 16 | 11.0M | 4.81 | 22.07 | - |
| | 28 | 36.5M | 4.17 | 20.50 | - |
| with Dropout | 16 | 2.7M | - | - | 1.64 |
| CapProNet+WideResNet (c=4) | 16 | 11.0M | 4.20 | 21.33 | - |
| | 28 | 36.5M | **3.64** | 19.98 | - |
| with Dropout | 16 | 2.7M | - | - | 1.58 |
| CapProNet+WideResNet (c=8) | 16 | 11.0M | **4.04** | **20.12** | - |
| | 28 | 36.5M | 3.85 | **19.83** | - |
| with Dropout | 16 | 2.7M | - | - | **1.54** |
| Densenet-BC k=12 [10] | 100 | 0.8M | 4.51 | 22.27 | 1.76 |
| CapProNet Densenet-BC k=12 (c=4) | 100 | 0.8M | 4.35 | 21.22 | 1.64 |
| CapProNet Densenet-BC k=12 (c=8) | 100 | 0.8M | **4.25** | **21.19** | **1.64** |

Table 2: The CapProNet results with Resnet50 and Resnet101 backbones for Single crop top-1/top-5 error rate on ImageNet validation set with image resolution of $224 \times 224$, as well as the comparison with original baseline results.

| Method | reported result[6] | our rerun | CapProNet (c=2) | CapProNet (c=4) | CapProNet (c=8) |
|---|---|---|---|---|---|
| Resnet50 | 24.8 / 7.8 | 24.09/7.13 | 23.282 / 6.8 | 23.265 / **6.648** | **23.203** / 6.78 |
| Resnet101 | 23.6 / 7.1 | 22.81 /6.67 | 22.192 / 6.178 | **21.89 / 6** | 21.9 / 6.01 |

successfully outperforms the original backbones on both Top-1 and Top-5 error rates. It is worth noting the gains are only obtained with the last layer of backbones replaced by the capsule project layer. We believe the error rate can be further reduced by replacing the intermediate convolutional layers with the capsule projections, and we leave it to our future research.

We also note that the CapProNet+X consistently outperforms the backbone counterparts with varying dimensions $c$ of capsule subspaces. In particular, with the WideResNet backbones, in most cases, the error rates are reduced with an increasing capsule dimension $c$ on all datasets, where the smallest error rates often occur at $c = 8$. In contrast, while CapProNet+X still clearly outperforms both ResNet and ResNet (pre-activation) backbones, the error rates are roughly at the same level. This is probably because both ResNet backbones have a much smaller input dimension $d = 64$ of feature vectors into the capsule projection than that of WideResNet backbone where $d = 128$ and $d = 160$ with 16 and 28 layers, respectively. This turns out to suggest that a larger input dimension can enable to use capsule subspaces of higher dimensions to encode patterns of variations along more directions in a higher dimensional input feature space.

To further assess the effect of capsule projection, we compare with the method that simply groups the output neurons into capsules without performing orthogonal projection onto capsule subspaces. We still use the lengths of these resultant "capsules" of grouped neurons to classify input images and the model is trained in an end-to-end fashion accordingly. Unfortunately, this approach, namely GroupNeuron+ResNet in Table 3, does not show significant improvement over the backbone network. For example, the smallest error rate by GroupNeuron+ResNet is 6.26 at $c = 2$, a small improvement

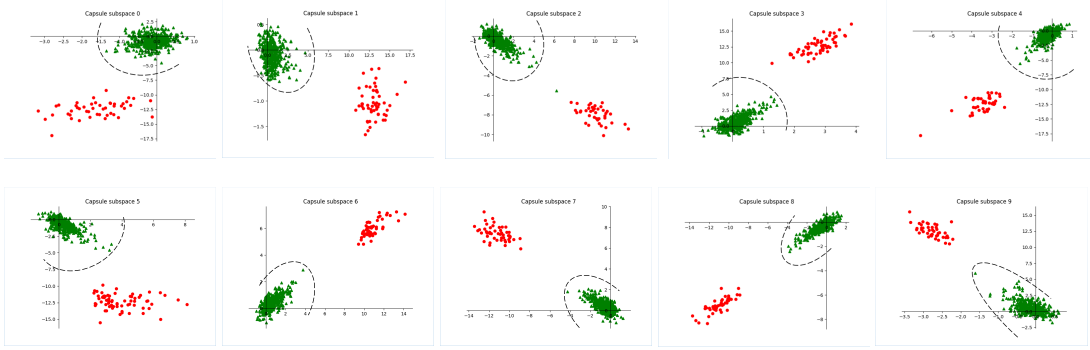

Figure 2: These figures plot the 2-D capsule subspaces and projected capsules corresponding to ten classes on CIFAR10 dataset. In each figure, red capsules represent samples from the class corresponding to the subspace, while green capsules belong to a different class. It shows red samples have larger capsule length (relative to the origin) than those of green samples. This validates the capsule length as the classification criterion in the proposed model. Note that some figures have different scales in two axes for a better illustration.

over the error rate of $6.41$ reached by ResNet110. This demonstrates the capsule projection makes an indispensable contribution to improving model performances.

When training on CIFAR10/100 and SVHN, one iteration typically costs $\sim 0.16$ seconds for Resnet-110, with an additional less than $0.01$ second to train the corresponding CapProNet. That is less than $1\%$ computing overhead. The memory overhead for the model parameters is even smaller. For example, the CapProNet+ResNet only has an additional $640 - 6400$ parameters at $c = 2$ compared with 1.7M parameters in the backbone ResNet. We do not notice any large computing or memory overheads with the ResNet (pre-activation) or WideResNet, either. This shows the advantage of CapProNet+X as its error rate reduction is not achieved by consuming much more computing and memory resources.

## 5.4 Visualization of Projections onto Capsule Subspaces

To give an intuitive insight into the learned capsule subspaces, we plot the projection of input feature vectors onto capsule subspaces. Instead of directly using $\mathbf{P}_l \mathbf{x}$ to project feature vectors onto capsule subspaces in the original input space $\mathbb{R}^d$, we use $(\mathbf{W}_l^T \mathbf{W}_l)^{-\frac{1}{2}} \mathbf{W}_l^T \mathbf{x}$ to project an input feature vector $\mathbf{x}$ onto $\mathbb{R}^c$, since this projection preserves the capsule length $\|\mathbf{v}_l\|_2$ defined in (2).

Figure 2 illustrates the 2-D capsule subspaces learned on CIFAR10 when $c = 2$ and $d = 64$ in CapProNet+ResNet110, where each subspace corresponds to one of ten classes. Red points

Table 3: Comparison between GroupNeuron and CapProNet with the ResNet110 backbone on CIFAR10 dataset. The best results are highlighted in bold for $c = 2, 4, 8$ capsules. It shows the need of capsule projection to obtain better results.

| c | GroupNeuron | CapProNet |
|---|---|---|
| 2 | 6.26 | **5.24** |
| 4 | 6.29 | **5.27** |
| 8 | 6.42 | **5.19** |

represent the capsules projected from the class of input samples corresponding to the subspace while green points correspond to one of the other classes. The figure shows that red capsules have larger length than green ones, which suggests the capsule length is a valid metric to classify samples into their corresponding classes. Meanwhile, the orientation of a capsule reflects various instantiations of a sample in these subspaces. These figures visualize the separation of the lengths of capsules from their orientations in classification tasks.

# 6 Conclusions and Future Work

In this paper, we present a novel capsule project network by learning a group of capsule subspaces for different classes. Specifically, the parameters of an orthogonal projection is learned for each class and the lengths of projected capsules are used to predict the entity class for a given input feature vector. The training continues until the capsule subspaces contain input feature vectors of corresponding classes or the back-propagated error vanishes. Experiment results on real image datasets show that the proposed CapProNet+X could greatly improve the performance of backbone network without incurring large computing and memory overheads. While we only test the capsule projection as the output layer in this paper, we will attempt to insert it into intermediate layers of backbone networks as well, and hope this could give rise to a new generation of capsule networks with more discriminative architectures in future.

## Acknowledgements

L. Zhang and M. Edraki made equal contributions to implementing the idea: L. Zhang conducted experiments on CIFAR10 and SVHN datasets, and visualized projections in capsule subspaces on CIFAR10. M. Edraki performed experiments on CIFAR100. G.-J. Qi initialized and formulated the idea, and prepared the paper.

## Footnotes

[2] A projection matrix $\mathbf{P}$ for a subspace $\mathcal{S}$ is a symmetric idempotent matrix (i.e., $\mathbf{P}^T = \mathbf{P}$ and $\mathbf{P}^2 = \mathbf{P}$) such that its range space is $\mathcal{S}$.

[3]https://github.com/tensorflow/models/tree/master/official/resnet

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
