[Reviews · NeurIPS 2018]

Reviewer 1



This paper introduces an alternative to CNN based architectures being inspired by the recently proposed capsule networks. The authors proposed to replace the last layer of ResNet variants by a capsule projection network, thereby getting promising results on the CIFAR and SVHN datasets. However, the motivation for using a capsule projection layer is unclear even though the technique is straightforward and easy to implement with minor computational overhead. The main idea of the capsule projection layer is to project the input feature vector to some learnt capsule subspaces (one for each class in classification setting), which are then used to distinguish between the different classes in classification. The authors also show that this projection technique leads to computation of gradients which are orthogonal to the learnt subspace, enabling discovery of novel characteristics leading to improvement of the learnt subspace. They have also shown interesting visualizations indicating separability of the samples for every classes. The quantitative results in this paper are encouraging when compared with the baselines used. Strengths : 1. Straightforward idea which is easy to implement with minimal computational overhead. 2. Promising experimental results with interesting visualizations. Weaknesses: 1. The motivation or the need for this technique is unclear. It would have been great to have some intuition why replacing last layer of ResNets by capsule projection layer is necessary and why should it work. 2. The paper is not very well-written, possibly hurriedly written, so not easy to read. A lot is left desired in presentation and formatting, especially in figures/tables. 3. Even though the technique is novel, the contributions of this paper is not very significant. Also, there is not much attempt in contrasting this technique with traditional classification or manifold learning literature. 4. There are a lot of missing entries in the experimental results table and it is not clear why. Questions for authors: Why is the input feature vector from backbone network needed to be decomposed into the capsule subspace component and also its component perpendicular to the subspace? What shortcomings in the current techniques lead to such a design? What purpose is the component perpendicular to the subspace serving? The authors state that this component appears in the gradient and helps in detecting novel characteristics. However, the gradient (Eq 3) does not only contain the perpendicular component but also another term x^T W_l^{+T} - is not this transformation similar to P_l x (the projection to the subspace). How to interpret this term in the gradient? Moreover, should we interpret the projection onto subspace as a dimensionality reduction technique? If so, how does it compare with standard dimensionality reduction techniques or a simple dimension-reducing matrix transformation? What does "grouping neurons to form capsules" mean - any reference or explanation would be useful? Any insights into why orthogonal projection is needed will be helpful. Are there any reason why subspace dimension c was chosen to be in smaller ranges apart from computational aspect/independence assumption? Is it possible that a larger c can lead to better separability? Regarding experiments, it will be good to have baselines like densenet, capsule networks (Dynamic routing between capsules, Sabour et al NIPS 2017 - they have also tried out on CIFAR10). Moreover it will be interesting to see if the capsule projection layer is working well only if the backbone network is a ResNet type network or does it help even when backbone is InceptionNet or VGGNet/AlexNet.

Reviewer 2



This paper proposes CapProNet, which uses a (capsule) vector rather than a neuron to represent class labels. The feature vector of an input sample is projected to a parametrized subspace. The idea is inspired by the capsule network, but the implementation is much simper and the computational cost is very small. Experiments on CIFAR10, CIFAR100 and SVHN with different model architecture show the proposed model consistently improves the performance. In general, the proposed model is simple and effective and the experiments are thorough. The authors claim the computational overhead of the proposed model is small, but why you experiment your model only on small scale dataset? It will be very interesting to see how the model perform on large scale dataset (i.e., imagenet).

Reviewer 3



This paper adopts the capsule vector idea from the capsule network and proposes the idea of dividing the feature space into multiple orthogonal subspace, one for each classification target category. Given a feature vector, this model first project it onto multiple orthogonal subspace and then use the 2 norm of the image vector to calculate softmax probability. Experiments show promising accuracy enhance. The idea is valuable, but it may have discarded some core thoughts of original capsule network. Pros: 1. It is a novel idea to project feature vector into orthogonal subspace. It’s motivated by the vector representation and length-to-probability ideas from capsule network, and the author did one more step to uncover new things. This is an interesting idea with a good math intuition. The proposed orthogonal subspace projection provides a novel and effective method to formalize the principled idea of using the overall length of a capsule. 2. On several image classification benchmarks, the CapProNet shows improvement (10%-20% reduced error rate) than state-of-art ResNet by incorparating the CapProNet as the last output layer. Additionally, the ResNet with CapProNet caused only <1% and 0.04% computing and memory overhead during model training than original ResNet. The CapProNet is highly extensible for many existing network. It might become a useful method that can be used in a more general setup. 3. The way the authors handle inverse matrix gradient propagation is interesting. 4. The presentation of the paper is clear; e.g., the presentation of visualization results of projections onto capsule subspaces in Section 5.2 is good. Cons: 1. It may be arguable if models presented in this paper should be called a capsule network: only the neuron group idea is inherited from the capsule network paper and other valuable core thoughts are discarded. For example, the capsule network introduces dynamic routing which grab confident activation through coincidence filtering, and different levels of capsules can learn part-whole hierarchy. However in this paper the second last layer is a single feature vector, which is bound to diverge from some core thoughts because it is likely we cannot find pattern’s that agrees during votes. 2. While the way the authors handle inverse matrix gradient propagation is interesting and it does not harm the gradient that is back propagated toward lower layers, I wonder whether it’s efficient enough to perform the m*n route by agreement scheme proposed by the original capsule paper. 3. The comparison in Table 1 doesn't include the latest state-of-art models on these image classification benchmarks, i.e., DenseNet (Huang et al., 2017) which achieves better results than CapProNet/ResNet on CIFAR-10/100. I think it may be more convincing to perform CapProNet experimants based on DenseNet or other latest state-or-art models. The comparison in Table 2 was not detailed enough. There should be more description in this part since CapProNet is very similar to "GroupNeuron" in the surface form. More detailed and analytical discussion between CapProNet and "GroupNeuron" would be helpful. 4. Still the paper can be better written, e.g., improving the positions/arrangement of tables and fixing exiting typos.